🔓 | **Open Peer Review** | Microbial Ecology | Research Article

# Effects of applying locust frass on the soil properties and microbial community in a peach orchard

Ji'ang Nie,[1] Hao Chen,[2] Yongdong Wang,[1] Dapeng Zhang,[3] Yao Wang,[2] Zheng Gao,[1] Ningxin Wang[2]

**ABSTRACT** Insect frass, a by-product of insect cultivation, represents an emergent category of agricultural waste. How to develop and make full use of this waste is a major issue to be addressed. In this study, the locust (*Locusta migratoria*) frass was used as an organic fertilizer in a peach orchard, and its effects on soil were investigated, including physicochemical properties and microbial community over three consecutive years. Compared to chemical fertilizer (CF), the application of locust frass (LF) enhanced the alpha diversity and network complexity of the soil microbial community. Furthermore, locust frass application led to an enrichment of functional microbial groups associated with the nitrogen cycle and enhanced the contents of nitrogen in the soil. *Bacillus* was identified as keystone microbes that discriminated between locust frass and chemical fertilizer treatments, and promoted the growth of Chinese cabbage (*Brassica campestris* L.) in pot experiments. The abundance of *Bacillus* diminished with successive applications of chemical fertilizers, whereas its prevalence remained consistent in the soil with application of locust frass. Importantly, locust frass application led to an enrichment of functional microbial groups associated with the nitrogen cycle and enhanced the contents of nitrogen in the soil. Structural equation modeling (SEM) suggests that nitrite respiration after locust frass application had a negative effect on soil total nitrogen content and that total nitrogen affected the entire taxa by affecting the rare taxa. Our results provided a theoretical basis for the application of locust frass as organic fertilizer.

**IMPORTANCE** Insect frass is a new type of organic fertilizer. The effects of some insect frass on soil have been investigated. However, the effect of locust frass (LF) (a by-product of the rearing process of this major edible insect) as an organic fertilizer is not well known. This study investigates the effects of locust frass on soil properties and the microbial community, providing valuable guide on its use as an organic fertilizer in agricultural production.

**KEYWORDS** *Locusta migratoria*, locust frass, soil microbial community, soil properties, peach orchard

The overuse of chemical fertilizers (CFs) leads to increased energy and non-renewable resources consumption while causing soil nutrient imbalance and a decline in soil microbial community diversity, which is detrimental to soil health (1–3). In recent years, organic fertilizers have been described as a potential alternative to chemical fertilizers. Some studies showed that organic fertilizers improved soil quality and increased plant growth (4–6).

Agriculture waste includes leaves, rotten fruits, and other wastes, which are often disposed of by burning or piling up in the field (7). This practice results in greenhouse gas emissions and adversely affects the environment. Utilizing insects for the recycling of this agricultural waste may offer a viable solution. The cultivation of insects using agricultural waste transforms low-value by-products into high-value proteins for food

**Peer Reviewer** Laith Khalil Tawfeeq Al-Ani, University of Baghdad, Baghdad, Iraq

Address correspondence to Ningxin Wang, nxwang@sdau.edu.cn, or Zheng Gao, gaozheng@sdau.edu.cn.

The authors declare no conflict of interest.

See the funding table on p. 16.

and animal feed (8). Besides the value of the insects themselves, insect frass, a by-product of farmed insects, is considered a new organic fertilizer with economic and environmental advantages (9). Insect frass is a mixture of insect excrement, undigested organic waste, and shed exoskeletons with a high organic matter and nutrient content, including nitrogen, phosphorus, and potassium (10). Insect frass plays a key role in this circular farming process, as it facilitates the recycling and reusing of agricultural waste to restore fertility to the soil, thus maintaining agricultural production.

Soil microorganisms are an important part of the soil ecosystem, affecting nutrient cycling, pollutant degradation, soil-borne diseases, and plant growth and immunity (11, 12). Watson et al. (13) found that adding black soldier fly frass, buffalo worm frass, or mealworm frass to soil could increase bacteria and archaea's 16S rRNA gene copy numbers. Furthermore, the application of black soldier fly frass, house cricket frass, or mealworm frass significantly altered the bacterial community structure in the rhizosphere soil of Brussels sprouts (14). Additionally, the application of black soldier fly frass increased the alpha diversity of bacterial and fungal communities in the rhizosphere soil of maize and revealed significant differences in these communities compared to those in soils without the frass application (15).

The application of insect frass can change soil physicochemical properties and enzyme activities to improve soil quality (9). It can also increase the contents of nitrogen, phosphorus, and potassium (13, 16). Additionally, some studies have found that the application of insect frass increased soil enzyme activities, such as dehydrogenase and β-glucosidase (9, 17, 18).

Locusts (*Locusta migratoria*) are the main edible Orthoptera group (19). Under the European Union's regulations, locusts are one of the entomological species authorized for human consumption, in addition to house crickets and mealworms (20). Large numbers of locusts were cultivated for food and animal feed (21). Therefore, more locust frass (LF) was produced. However, the effects of applying locust frass on soil properties and microbial communities remain unknown. Thus, it is imperative to conduct studies on the ecological impact of locust frass on soil, particularly its utility as an organic fertilizer.

The application of fertilizers is essential for many orchards that have low soil nutrient content (22). This study aimed to (i) compare soil microbial community structure and physicochemical properties under locust frass and chemical fertilizers, (ii) identify keystone microbes post-fertilization, and (iii) analyze their effects and relationships with soil properties. The findings support locust frass utilization for soil quality, soil health conservation, and sustainable agriculture.

## MATERIALS AND METHODS

### Study site and sample collection

The study was conducted in Mazhuang Town, Tai'an City, Shandong Province, China (36°00′11″N, 116°96′44″W). This area has a temperate monsoon climate and an average annual sunshine of 2,627.1 days, with a sunshine rate of about 58%. The average annual temperature is 13°C, ranging from −2.6°C in January to 26.4°C in July. The average frost-free period lasts for 195 days. The average annual precipitation is 697 mm, and the primary source of irrigation is groundwater. The orchard soil is sandy loam with basic physical and chemical properties (organic matter 3.56%, available nitrogen 114.9 mg/kg, available potassium 91 mg/kg, available phosphorus 75.9 mg/kg).

*Prunus persica* Batsch cv. Jinqiuhongmi is a peach tree variety. Two treatments, CF and LF, were applied consecutively for 2 and 3 years since planting the peach tree in 2016. All fertilizers were applied in furrows, with an average of 400 g per plant per year for chemical fertilizer (produced by Tai'an Runfeng Agricultural Technology Development Co., Ltd., with total nutrients ≥50% and trace elements ≥2%; the N/P/K ratio was 10:20:10) and 40 kg per plant per year for locust frass. The application quantity of chemical fertilizer was established in accordance with the guidelines provided by the manufacturer. Baldi et al. (23) suggested that the advised application of animal manure

in peach orchards is approximately 100 times that of chemical fertilizers. Consequently, locust frass application is also executed at a ratio of 100 to that of chemical fertilizer application. Initial soil was collected before planting the peach trees in 2016 as blank control, and locust frass was also taken before application in the soil. Soil samples were collected in October of 2018 and 2019 after the application of locust frass as fertilizer for 2 and 3 years, and peach trees of uniform growth and health were selected as test subjects, with five trees selected from each group. Within the drip line of the tree, two points were selected below the main branch in the east-west direction. The root-zone soil was taken at a depth of 30 cm after removing the topsoil. Then, about 800 g was taken in a self-sealing bag by quartering method and brought back at a low temperature to maintain the physical and chemical properties of the soil. The roots were removed from the root-zone soil and brought back in a self-sealing bag at a low temperature (4°C) for elution to obtain rhizosphere soil to extract the total soil DNA. Rhizosphere soil was stored at −80°C until DNA extraction, and the rest of the samples were processed immediately for physicochemical analysis and isolation of microorganisms. Five soil samples were collected as five replicates for subsequent analysis and microbial DNA extraction.

## Determination of soil physicochemical characteristics and enzyme activities

Soil pH was measured using a pH electrode at a soil/water ratio of 1:1 (wt:vol) (24). Soil Organic Material (OM) was first oxidized by $K_2Cr_2O_7$ and heated to 180 °C for 5 minutes, and then the excess $K_2Cr_2O_7$ was determined by titration with a standard 0.2 mol $L^{-1}$ $FeSO_4$ (25). Soil total nitrogen (TN) was analyzed using the Kjeldahl method (26). The soil samples were subjected to high-temperature digestion at 350°C with concentrated $H_2SO_4$ and catalysts to transmute both organic and inorganic nitrogen into $NH^{4+}$. $NH^{4+}$ in the digest was executed via acidimetric titration subsequent to alkaline distillation. Soil available nitrogen (AN) was determined using the alkali-diffusion method (27). The soil samples were subjected to a sodium hydroxide treatment to liberate the ammonia. The ammonia absorbed by the boric acid was determined by titration with a stand-ard acid using a methyl red-methylene blue indicator. Soil available phosphorus (AP) was measured by the Olsen method (28). Phosphorus in the soil was extracted with sodium bicarbonate solution and then added to molybdate for colorimetry. Soil available potassium (AK) was determined by the ammonium acetate extraction flame photometry method (29). Potassium in the soil was extracted with ammonium acetate solution and then determined by flame spectrophotometry. Urease (URE) activity was determined using the indophenol-blue colorimetry method (30). The soil samples were incubated with an aqueous urea solution, and ammonium was extracted with KCl and HCl, followed by the indophenol-blue colorimetric method for the determination of $NH_4^+$. Catalase (CAT) activity was determined using the potassium permanganate titration method (31). The soil samples reacted with hydrogen peroxide, and the excess peroxide was measured by titration with potassium permanganate. Sucrase (SUC) activity was determined using the 3,5-dinitrosalicylic acid colorimetric method (32). The soil samples were incubated with buffered sucrose solution and toluene for 24 hours at 37°C, and then the 3,5-dinitrosalicylic acid was added to measure the absorbance at optical density $(OD)_{540}$. Phosphatase (PHO) activity was determined using the p-nitrophenyl phosphate disodium colorimetry method (33). The soil samples were incubated with buffered sodium p-nitrophenyl phosphate solution and toluene for 1 hour at 37°C, after which the absorbance was measured at $OD_{400}$.

## Isolation and identification of microorganisms

A 100 µL of peach tree rhizosphere soil solution serially diluted $10^3$ to $10^6$ times with sterile water was spread on plates of Luria-Bertani (LB) medium and was incubated at 28°C. Different strains were selected from each plate and were purified by the streak method on LB medium. After purification, the strains were stored at −20°C using glycerol.

For molecular identification, universal primers 27F (5′-AGAGTTTGATCCTGGCTCAG-3′) and 1492R (5′-GGTTACCTTGTTACGACTT-3′) were used to amplify the 16S rRNA genes. The final volume of amplification reaction solution was 25 µL, containing 18.35 µL of ddH$_2$O, 2.5 µL of 10 buffer, 2 µL of dNTP, 0.5 µL of each primer, 0.15 µL of Taq polymerase, and 1 µL of template solution. The PCR conditions were as follows: initial denaturation at 95℃ for 10 minutes, thermal denaturation at 95℃ for 40 seconds, annealing at 53℃ for 40 seconds, extension at 72℃ for 90 seconds for a total of 35 cycles, and finally, extension at 72℃ for 10 minutes. Sequence homology using BLAST in the NCBI database (https://www.ncbi.nlm.nih.gov/) was carried out for the obtained 16S rRNA gene sequences.

## Pot experiment

Considering the perennial nature and long growth cycle of the peach tree, we selected the common vegetable crop Chinese cabbage (*Brassica campestris* L.) as a suitable host for testing the three bacteria strains isolated from the peach tree rhizosphere and conducting plant growth experiments. Chinese cabbage in a medium consists of soil and vermiculite at the ratio of 1:1. A 32-hole seedling tray was selected, and two to three seeds were planted in each tray. At the time of exposure of the first main leaf, the bacterial solutions of *Bacillus aryabhattai*, *Bacillus megaterium*, and *Bacillus paramycoides* (OD$_{600}$ = 0.5) were inoculated into the wells at 10 mL each. A blank control was inoculated with an equal volume of LB medium. Plant growth was measured after 45 days. Plant height, root length, shoot fresh weight, root fresh weight, shoot dry weight, and root dry weight were determined, and three replicates were set up.

## DNA extraction and high-throughput sequencing

The total DNA of rhizosphere soil was extracted using an OMEGA (D5625-01) kit, and the total extracted DNA was sent to the Shanghai Majorbio Bio-Pharm Technology Co., Ltd (Shanghai, China). The Illumina MiSeq platform was selected to sequence the DNA from different rhizosphere soil samples by high-throughput sequencing. Universal bacterial primers 338F (5′-ACTCCTACGGGAGGCAGCAG-3′) and 806R (5′-GGACTACHVGG GTWTCTAAT-3′) were used for the amplification of the V3-V4 regions of the bacterial 16S rRNA gene. Universal fungal primers ITS1F (5′-CTTGGTCATTTAGAGGAAGTAA-3′) and ITS2R (5′-GCTGCGTTCTTCATCGATGC-3′) were used for the amplification of the fungal internal transcribed spacer (ITS) region.

## Analysis of the sequencing data

After high-throughput sequencing, the raw sequence data were preliminarily processed using USEARCH v10.0.240 software (34). The barcodes and primer sequences were removed, and sequence splicing was merged, trimmed, and filtered. The sequences were clustered by operational taxonomic units (OTUs) using a 97% identity threshold. Furthermore, a representative sequence for each OTU was randomly selected and aligned with the SILVA database 138 (http://www.arb-silva.de/). The bacterial and fungal sequences were identified and classified as 3,629 and 779 OTUs, respectively. To compare the diversity of the microbial communities, all samples were rarefied to the same depth based on the lowest sequence number using QIIME v.1.9.0. A total of 18,338 sequences for each sample of bacteria and 44,603 sequences for each sample of fungi were obtained.

When the relative abundance of OTUs in all samples was more than 0.1%, they were defined as "abundant taxa," while those with a relative abundance of less than 0.01% were defined as "rare taxa" (35, 36). Among the bacteria, 2,164 OTUs were rare taxa, and 158 OTUs were abundant taxa. Among the fungi, 435 OTUs were rare taxa, and 89 OTUs were abundant taxa.

## Statistical analysis

The normality of the data was assessed using the Shapiro-Wilk normality test in the R package "MVN" (37). *T*-test and one-way analysis of variance (ANOVA) with Duncan's HSD test were performed using the R software. BLAST was used to compare the sequences of isolated *Bacillus* and OTUs. The R package "DESeq2" was used to calculate the differential OTUs between LF and CF treatments (38). The potential ecological function changes of soil bacterial microbial communities were evaluated using FAPROTAX v.1.2.1 (39). Fungal function guild was annotated using FUNGuild (40). The alpha diversity index, analysis of similarities (ANOSIM), and distance matrices were calculated using the R package "vegan" (41), while Bray-Curtis metrics were applied to the soil microbial community. The R package "igraph" was used to visualize the correlation network (42). Associations were determined by Spearman's correlation coefficient. Associations with $P < 0.05$ and correlation coefficients above 0.4 were retained. The correlation network properties were calculated using the R package "ggClusterNet" (43). Random forest-based models were built in the R package "randomForest" to define keystone microbes to differentiate bacterial communities (40). The microbial source-tracking method fast expectation-maximization microbial source tracking (FEAST) was performed using the R package "FEAST" to track the potential sources of the microbial community in the soil after locust frass application (44). The Mantel test was performed using the R package "linkET" to analyze the relationship between microbial communities and soil properties (45). Based on the results of the correlation analysis, crucial soil drivers were extracted, and the structural equation model (SEM) analysis was conducted using the R package "piecewiseSEM" (46).

## RESULT

### Structure and function of the soil microbial community

This study observed no significant difference in peach tree height after 2 to 3 years of locust frass application compared to chemical fertilizer, and no disease outbreaks were noted in either treatment. Then, high-throughput sequencing was used to analyze the soil microbial community structure post-fertilizer application. The phylum-level compositions of the bacterial and fungal entire, abundant, and rare taxa were similar (Fig. S1a [supplemental material is found at https://doi.org/10.6084/m9.figshare.28471364]). Proteobacteria (54.21%, 65.95%, 26.70%) and Acidobacteriota (13.59%, 10.58%, 14.86%) were the dominant phyla for the bacterial entire, abundant, and rare taxa, respectively. Similarly, Ascomycota (46.23%, 44.54%, 62.16%) and Basidiomycota (39.48%, 40.77%, 10.48%) were the dominant phyla for the fungal entire, abundant, and rare taxa, respectively. The entire and abundant taxa of bacteria and fungi showed a similar genus-level composition, which differed considerably from that of the rare taxa (Fig. S1b at https://doi.org/10.6084/m9.figshare.28471364). *Steroidobacter* (14.20%, 24.23%) and *Sphingomonas* (5.52%, 8.29%) were the dominant genera for the bacterial entire and abundant taxa, respectively, while *Gemmatimonas* (2.33%) and *Blrii41* (1.99%) were the dominant genera for the bacterial rare taxa. Likewise, *Tausonia* (33.65%, 35.70%) and *Mortierella* (12.39%, 13.45%) were the dominant genera for the fungal entire and abundant taxa, respectively, and *Mortierella* (5.47%) and *Chrysosporium* (3.48%) were the dominant genera for the fungal rare taxa.

Contrary to the top 10 phyla, the top 10 genera difference between LF and CF treatments emerged only after 2 years of continuous application (Fig. 1a and b). In order to further analyze the differences in soil microbial communities after the application of LF, DESeq2 was used to analyze the differences in soil microbial communities at the OTU level (Fig. 1c). The 2-year treatment resulted in 210 and 61 differential bacterial and fungal OTUs, respectively. After 3 years' treatment, there were 79 and 61 differential bacterial and fungal OTUs, respectively. There were 32 and 14 differential bacterial and fungal OTUs, respectively, that differed in both the 2- and 3-year treatments. The bacterial community showed more variation across time than the fungal community.

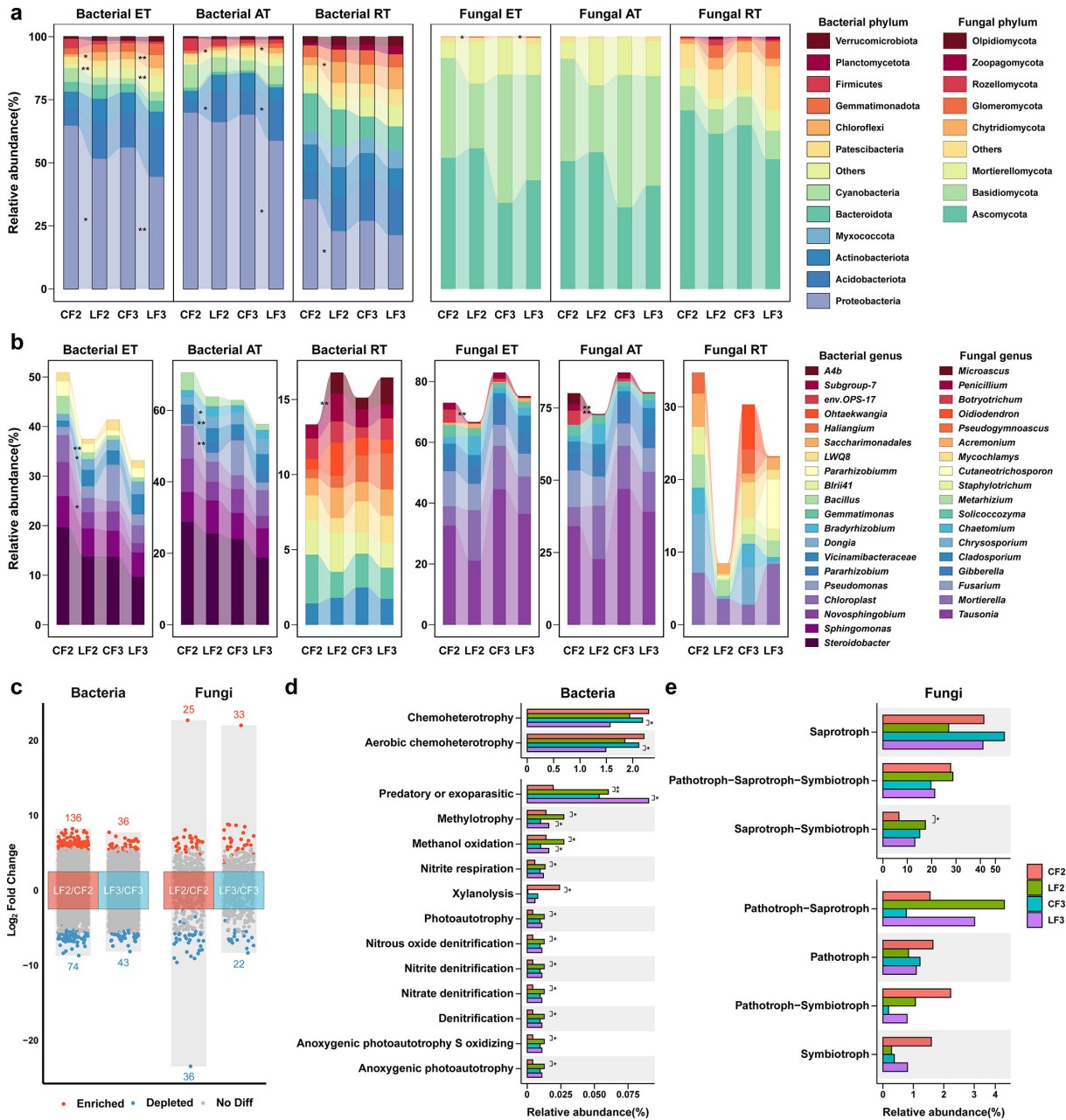

**FIG 1** Effects of different fertilization treatments on microbial community and function. Phylum-level (a) and genus-level (b) distribution of microbial communities. The top 10 phyla and genera with the highest average relative abundance for each treatment are presented. (c) Number of differential OTUs between the two sample groups. (d) Metabolic and ecological functions of bacteria based on FAPROTAX. (e) Functional guilds of fungi based on FUNGuild. The statistical significance of the differences between treatments within the same year was evaluated using the $t$-test, with asterisks representing the level of significance (***, $P < 0.001$; **, $P < 0.01$; *, $P < 0.05$). Five replicates were set up. ET, entire taxa; AT, abundant taxa; and RT, rare taxa.

A functional annotation analysis of prokaryotic taxa (FAPROTAX) was performed to generate predicted functional profiles based on soil bacterial community composition (Fig. 1d). After two consecutive years of treatment, compared to the CF treatment, the LF treatment enriched functional groups related to nitrogen and carbon metabolism, including methylotrophy, methanol oxidation, nitrite respiration, nitrous oxide denitrification, nitrite denitrification, nitrate denitrification, and denitrification. After three consecutive years of treatment, compared to the CF treatment, the LF treatment significantly enhanced methylotrophy and methanol oxidation. Although the nitrogen metabolism functional groups seemed to be numerically higher in the LF than in the CF

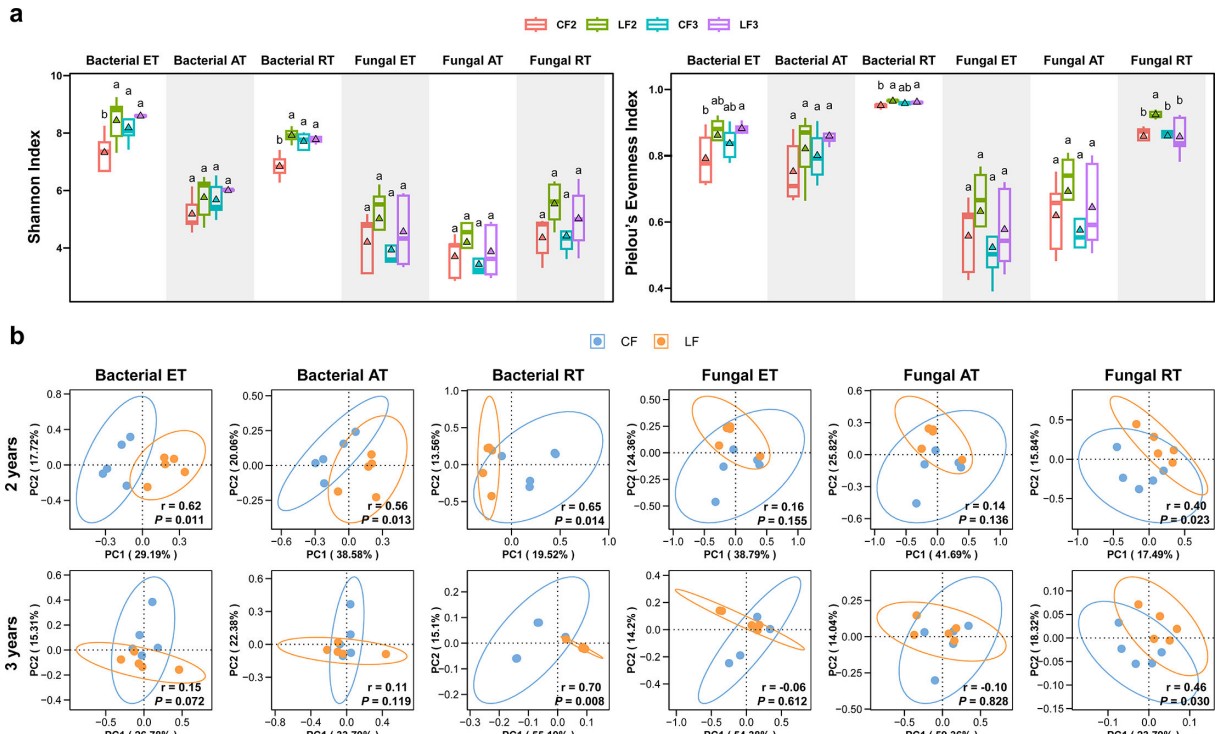

**FIG 2** Diversity of the soil microbial community. (a) Shannon index and Pielou's evenness index of microbial communities. The triangle represents the average of each group. Different labeled letters indicate significant differences between different treatments according to one-way ANOVA with Duncan's HSD test (P < 0.05). Five replicates were set up. (b) PCoA of microbial communities under different fertilization treatments after two (a) and three (b) consecutive years of treatment based on Bray-Curtis distance using ANOSIM. ET, entire taxa; AT, abundant taxa; and RT, rare taxa.

treatment, the difference was not statistically significant. FUNGuild was also used to infer fungal functional guilds (Fig. 1e). Seven main trophic modes (Pathotroph, Saprotroph, Symbiotroph, Saprotroph-Symbiotroph, Pathotroph-Symbiotroph, Pathotroph-Saprotroph, and Pathotroph-Saprotroph-Symbiotroph) were detected in the LF and CF treatments. Only Saprotroph-Symbiotroph was significantly enriched in the LF treatment compared to the CF treatment after two consecutive years of treatment.

## Diversity of the soil microbial community

The Shannon index and Pielou's evenness index for both bacterial and fungal communities in the LF treatments were higher than in the CF treatments, but the differences were not statistically significant after 3 years of continuous application (Fig. 2a). After two consecutive years of treatment, the Shannon index of the bacterial entire taxa, as well as the Shannon index and Pielou's evenness index of the bacterial and fungal rare taxa in the LF treatment, was found to be significantly higher than those in the CF treatment (P < 0.05). After 3 years of continuous application, no statistically significant differences were observed for the Shannon diversity index and Pielou's evenness index computed for CF and LF. In addition, the Shannon index of the bacterial entire taxa, as well as the Shannon index and Pielou's evenness index of the bacterial and fungal rare taxa, was significantly lower after 2 years' CF application than 3 years' CF application (P < 0.05).

The principal coordinate analysis (PCoA) plot demonstrated variations in beta-diversity among the entire, abundant, and rare taxa of bacterial and fungal communities across the samples (Fig. 2b). After two consecutive years of treatment, the entire, abundant, and rare taxa of bacterial communities were significantly distinct between CF and LF treatments (ANOSIM, P < 0.05). For fungi, only the rare taxa showed significant separation between treatments (ANOSIM, P < 0.05), whereas the entire and abundant taxa were both clustered. After three consecutive years of treatment, only the rare taxa

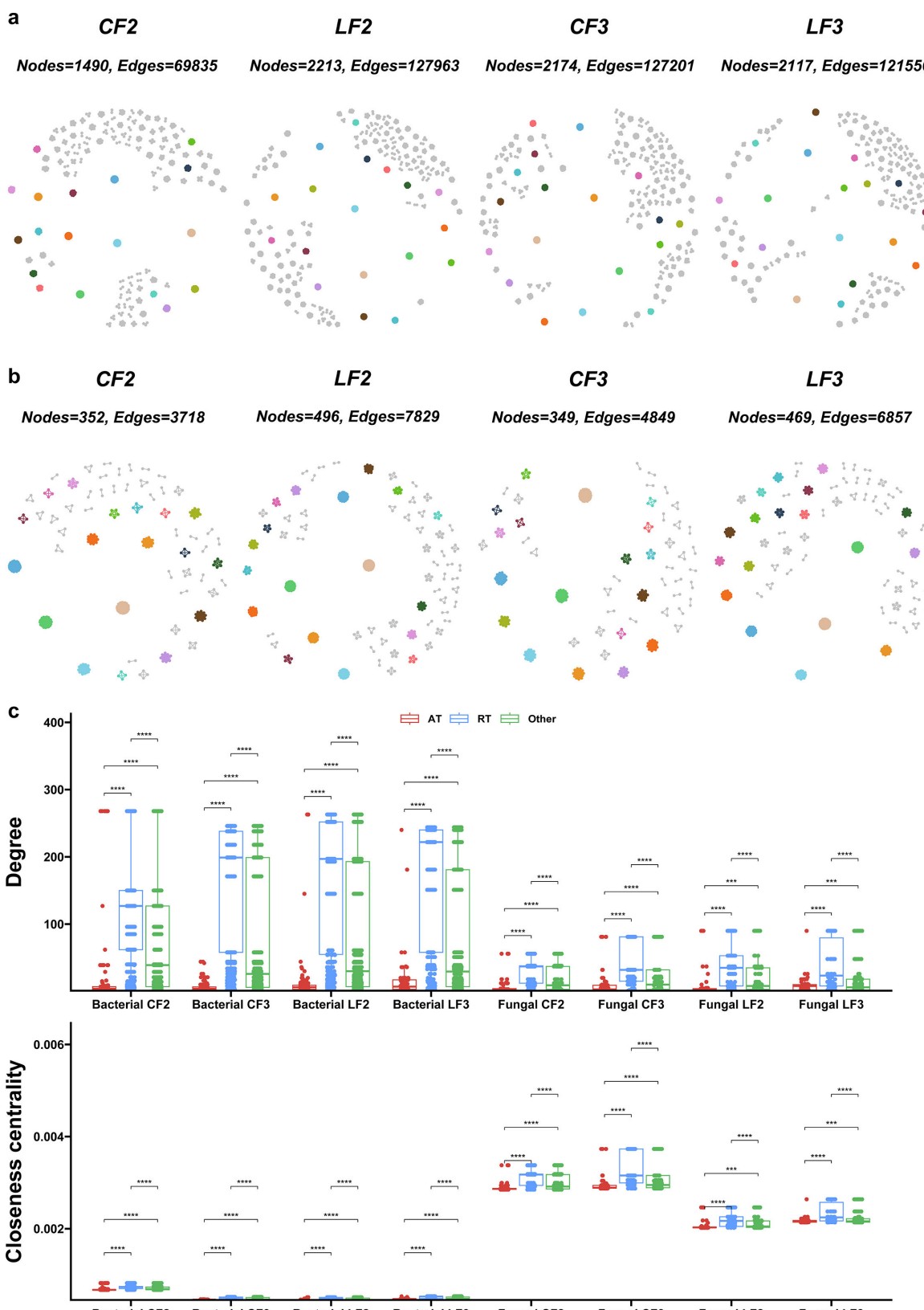

**FIG 3** Co-occurrence networks of soil bacterial (a) and fungal (b) communities under different fertilization treatments. Each node represents an OTU, coloring different modules. Five replicates were set up. (c) Node-level topological characteristics (degree and closeness centrality) of abundant, rare, and other taxa in microbial communities under different fertilization treatments. The statistical significance of the differences between treatments within the same year was evaluated using the *t*-test, with asterisks representing the level of significance (***, *P* < 0.001; **, *P* < 0.01; *, *P* < 0.05). ET, entire taxa; AT, abundant taxa; and RT, rare taxa.

of bacterial and fungal communities were significantly separated between CF and LF treatments (ANOSIM, $P < 0.05$), whereas the entire and abundant taxa were clustered. The results suggested that the different fertilization treatments induced significant differences in bacterial communities detectable after two consecutive years of treatment but not after 3 years. Additionally, no significant differences (except for the rare taxa) were observed in fungal communities after two and three consecutive years of treatment.

## Co-occurrence network of the soil microbial community

According to Spearman's correlations, co-occurrence networks of microbial communities were constructed at the OTU level to examine the interactions among microorganisms. After two consecutive years of treatment, the network of bacterial communities in the LF treatment had more nodes, edges, and clusters, and higher values of average degree and relative modularity than the CF treatment, indicating LF increased the complexity of network and degree of modularity (Fig. 3a, Table 1). Surprisingly, the network of bacterial communities in the LF treatment after three consecutive years was similar to the CF treatment. After two and three consecutive years of treatments, the network of fungal communities in the LF treatment had more nodes, edges, and clusters, and higher values of average degree and relative modularity than the CF treatment, showing that the LF treatment increased the complexity of network and degree of modularity (Fig. 3b, Table 1). The fungal networks showed less variation than the bacterial networks in their responses to the treatments, especially after 2 years. Moreover, the degree and closeness centrality of bacterial and fungal rare taxa were significantly higher than the abundant taxa, indicating that more information about the network may be transmitted through the rare taxa (Fig. 3c).

## Random forest machine learning model revealed keystone microbes and their plant-promoting capabilities

DESeq2, PCoA, and co-occurrence networks results showed that the bacterial community appeared to be more different after 2 years of continuous application of LF or CF. Random forest machine learning models were applied to define biomarkers to differentiate bacterial communities at the genus level after 2 years of continuous application of LF and CF (Fig. 4a). After 10-fold cross-validation with five repeats, 52, 25, and 18 bacterial genera were defined as biomarker taxa of the entire, abundant, and rare taxa, respectively (). *Bacillus* is the only genus that has been defined as a biomarker in the entire, abundant, and rare taxa (Fig. 4b). Therefore, *Bacillus* is defined as keystone microbes.

Based on random-forest-defined biomarkers, *Bacillus* was chosen for the plant growth promotion experiment. Three *Bacillus* strains, *B. aryabhattai*, *B. megaterium*, and *B. paramycoides*, were isolated from the rhizosphere soil of the peach trees. *B. aryabhattai* and *B. megaterium* exhibited a 100% congruence with OTU50 and OTU98. The relative abundances of *B. aryabhattai* and *B. megaterium* ranked as the second and

**TABLE 1** Topological properties of soil microbial network under different fertilization treatments

| | Bacteria | | | | Fungi | | | |
|---|---|---|---|---|---|---|---|---|
| Treatment | CF2 | LF2 | CF3 | LF3 | CF2 | LF2 | CF3 | LF3 |
| Edges | 69,835 | 127,963 | 127,201 | 121,556 | 3,718 | 7,829 | 4,849 | 6,857 |
| Positive edges | 69,666 | 127,599 | 126,790 | 121,015 | 3,698 | 7,808 | 4,831 | 6,817 |
| Negative edges | 169 | 364 | 411 | 541 | 20 | 21 | 18 | 40 |
| Vertices | 1,490 | 2,213 | 2,174 | 2,117 | 352 | 496 | 349 | 469 |
| Connectance | 0.063 | 0.052 | 0.054 | 0.054 | 0.060 | 0.064 | 0.080 | 0.062 |
| Average degree | 93.738 | 115.647 | 117.020 | 114.838 | 21.125 | 31.569 | 27.788 | 29.241 |
| Clusters | 123 | 145 | 154 | 143 | 56 | 66 | 51 | 65 |
| Centralization degree | 0.117 | 0.067 | 0.059 | 0.061 | 0.099 | 0.118 | 0.153 | 0.130 |
| Centralization closeness | 1.97E-04 | 6.97E-05 | 6.27E-05 | 6.64E-05 | 6.91E-04 | 5.99E-04 | 1.18E-03 | 7.04E-04 |
| Relative modularity | 13.992 | 20.428 | 20.922 | 20.372 | 3.926 | 4.789 | 2.952 | 3.698 |

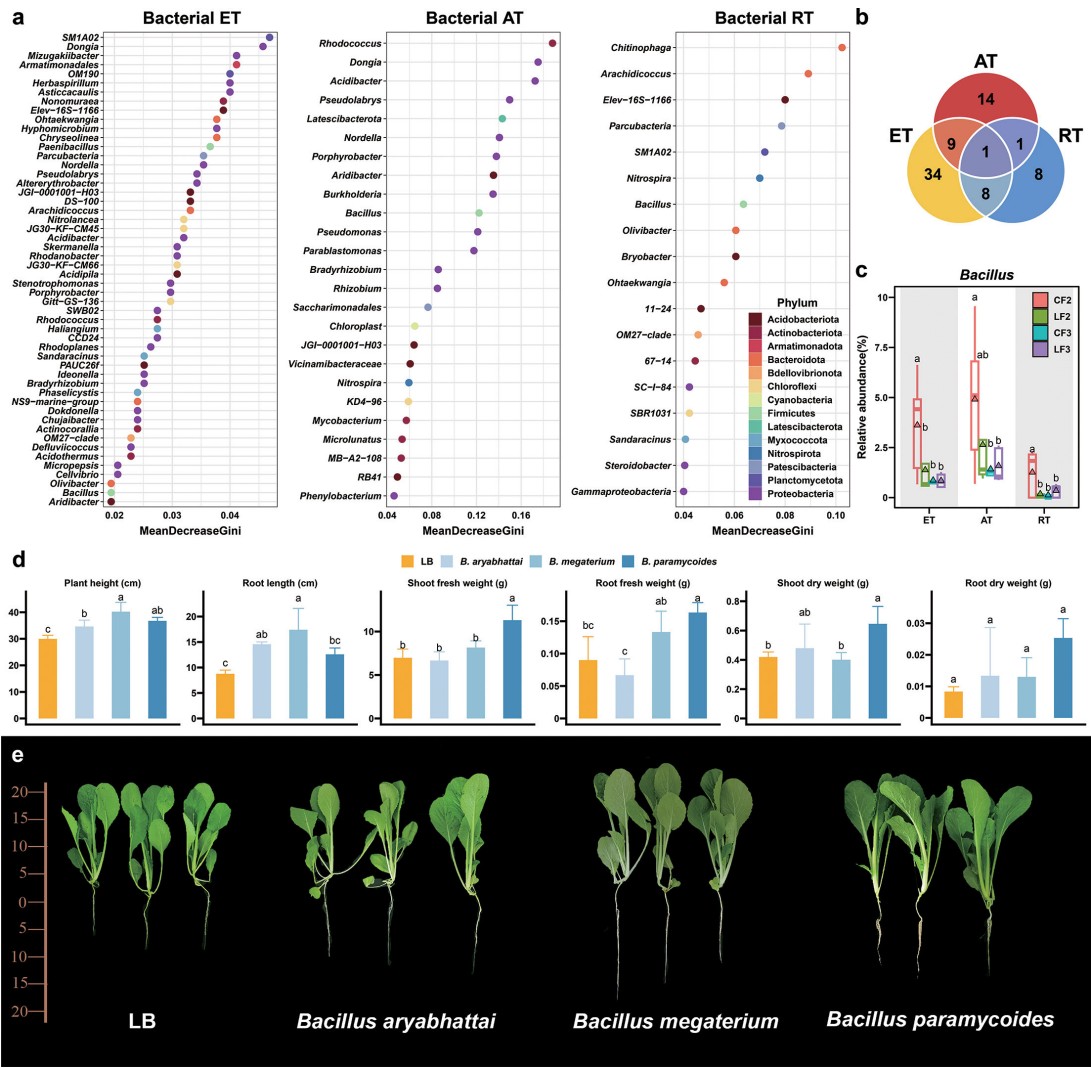

**FIG 4** Selection of keystone microbes and their growth-promoting phenotypes. (a) Biomarkers bacterial genera of ET, AT, and RT identified by random forest classification of bacterial communities after two consecutive years of locust frass or chemical fertilizer. (b) Venn diagrams presenting shared and unique biomarkers between ET, AT, and RT. (c) Relative abundance of *Bacillus* in ET, AT, and RT. Five replicates were set up. (d, e) Growth-promoting phenotypes of Chinese cabbage after inoculation with three strains of *Bacillus*. Three replicates were set up. The triangle represents the average of each group. Different labeled letters indicate significant differences between different treatments according to one-way ANOVA with Duncan's HSD test ($P < 0.05$). ET, entire taxa; AT, abundant taxa; and RT, rare taxa.

third most prevalent species within the genus *Bacillus*, contributing to 26.4% of the genus total abundance (). Pot-grown Chinese cabbage with the three *Bacillus* strains were inoculated. Plant height, root length, shoot fresh weight, root fresh weight, shoot dry weight, and root dry weight were measured to verify their growth-promoting ability (Fig. 4d and e). Cabbage inoculated with *B. aryabhattai* and *B. megaterium* showed a significant increase in plant height and root length, and cabbage inoculated with *B. paramycoides* had significant increase in plant height, root length, shoot fresh weight, root fresh weight, and shoot dry weight, as compared to the control inoculated with LB medium. All three *Bacillus* strains could promote the growth of Chinese cabbage and were beneficial bacteria with growth-promoting function. After two consecutive years of treatment, the relative abundance of *Bacillus* in the entire and rare taxa under the CF treatment was found to be significantly higher than those under the LF treatment ($P < 0.05$) (Fig. 4c). However, the relative abundance of *Bacillus* in the entire, abundant, and rare taxa after 3 years of application of CF was significantly lower than after 2 years

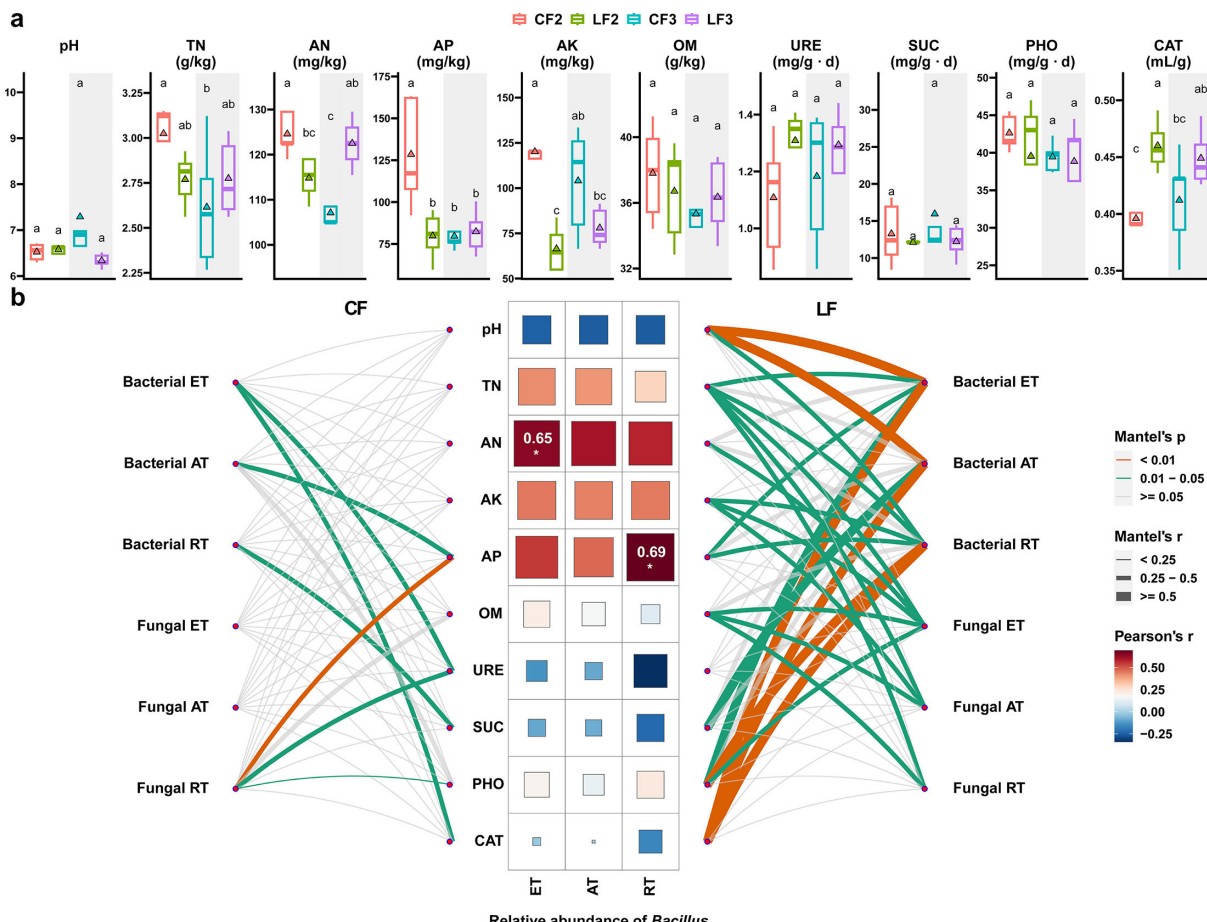

**FIG 5** Relationships between microbial community composition and soil properties. (a) Comparison of soil physicochemical and enzymatic activities under different fertilization treatments. Different labeled letters indicate significant differences between different treatments according to one-way ANOVA with Duncan's HSD test ($P < 0.05$). Five replicates were set up. (b) Mantel test showing the relationship between microbial communities and soil properties. Heatmap showing correlations between *Bacillus* and soil properties under chemical fertilizer treatment based on Pearson correlation analysis.

of application of CF to the soil ($P < 0.05$). According to our results, CF may reduce the abundance of beneficial bacteria in soil with increasing treatment duration.

## Relationships between microbial community composition and soil properties

Fast expectation-maximization microbial source tracking (FEAST) was used to track the potential sources of the bacterial and fungal community in the soil after the application of locust frass. Approximately 90% of the soil microbial community was originated from the soil, and less than 0.6% of the soil microbial community was originated from locust frass (Fig. S5 at https://doi.org/10.6084/m9.figshare.28471364), which suggested that the changes in soil microbial communities after locust frass treatment may be less influenced by the original locust frass microbiota. Hence, it is probable that the alterations in microbial communities are mainly driven by the changes in soil properties.

Different fertilization treatments had different effects on soil properties (Fig. 5a). After two consecutive years of treatment, soil AK and AP contents were significantly lower in the LF treatment than in the CF treatment ($P < 0.05$), while the difference was not significant after three consecutive years of treatment. After two consecutive years of treatment, the AN content was significantly lower in the LF treatment than in the CF treatment ($P < 0.05$). With increasing treatment time, the LF treatment significantly increased the content of soil AN compared to the CF treatment ($P < 0.05$), while TN, OM, and pH contents were not significantly different between the two treatments. For soil

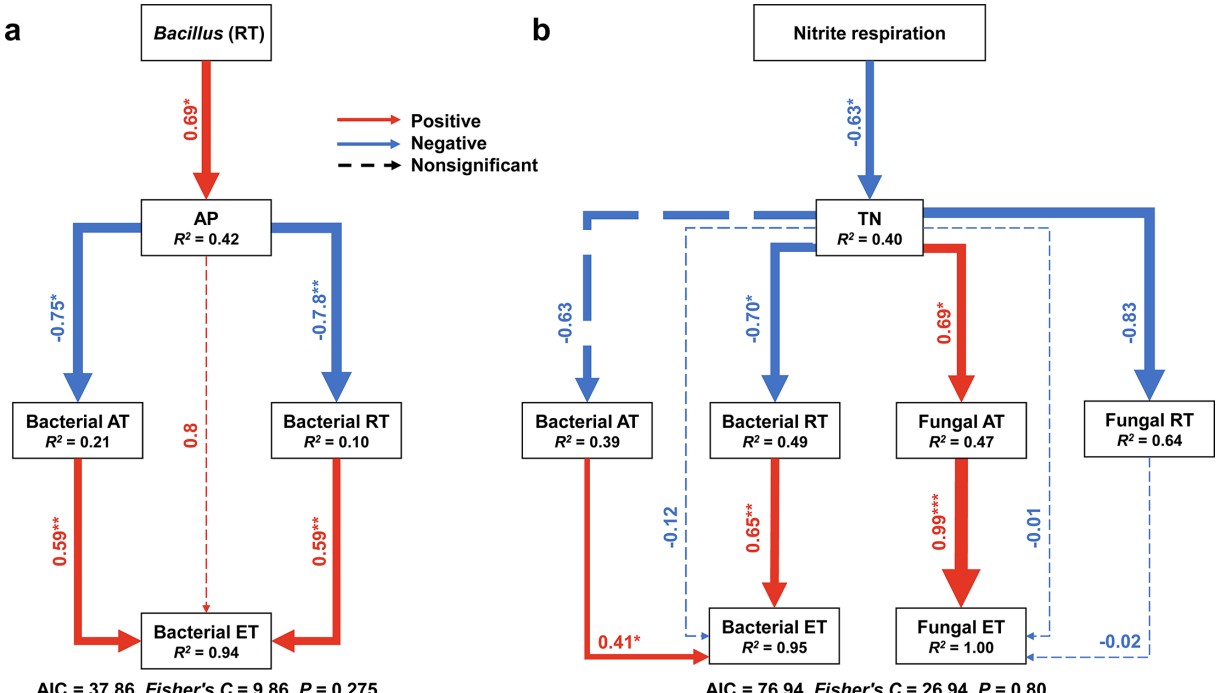

**FIG 6** SEM for the LF treatment (a) and CF treatment (b). The value above the SEM line represents the path coefficient, with asterisks representing the level of significance (***, $P < 0.001$; **, $P < 0.01$; *, $P < 0.05$). The red lines represent the positive path coefficient, the blue lines represent the negative path coefficient, and the dashed lines represent the nonsignificant path coefficient. The width of the arrow indicates the size of the standard path coefficient. Community structure is represented by the first axis of the PCoA.

enzymes, after two consecutive years of treatment, the activity of CAT was significantly higher in LF-treated soils than in CF ($P < 0.05$), while there were no significant differences in the activities of URE, SUC, and PHO between the two treatments.

To clarify the main driving factors of shaping the microbial community structure, Mantel tests were calculated based on microbial communities and soil properties (Fig. 5b). In the CF treatment, AP was the only soil physicochemical parameter significantly associated with microbial communities, and it was significantly correlated with the bacterial abundant taxa and fungal rare taxa. Compared to the CF treatment, more soil properties were significantly associated with the microbial community structure in the LF treatment. TN, AN, AK, AP, OM, SUC, PHO, CAT, and pH were significantly associated with the microbial community structure. Among them, TN has the highest correlation with the microbial taxa, and it was significantly correlated with all bacterial and fungal taxa (except for the bacterial rare taxa).

## Structural equation model analysis

Structural equation model (SEM) analysis was used to analyze the direct and indirect effects of the LF treatment and CF treatment on soil microbial communities. For the CF treatment, based on the results of the correlation analysis, AP may be a key physicochemical parameter that shaped the structure of microbial communities. In addition, *Bacillus*, identified as keystone microbes, had a significant positive correlation with AP in the bacterial rare taxa. SEM was constructed to analyze the direct and indirect effects of *Bacillus* in the bacterial rare taxa on AP and soil microbial communities (Fig. 6a). During the construction of the SEM, AP was not significantly correlated with the fungal entire, abundant, and rare taxa ($P > 0.05$). For model fit, all of the fungal taxa were removed from the model, and the final SEM fitted the data well (AIC = 37.86, Fisher's C = 9.86, $P = 0.275$). The results showed that *Bacillus* in the bacterial rare taxa had a positive effect on

the AP content. Additionally, AP indirectly affected the bacterial entire taxa by negatively affecting the bacterial abundant and rare taxa.

For the LF treatment, based on the results of the correlation analysis, TN may be a key physicochemical parameter that shaped the structure of microbial communities. Nitrite respiration was the only nitrogen metabolism-related functional group enriched in the LF treatment, which was significantly correlated with TN (). SEM was constructed to analyze the direct and indirect effects of nitrite respiration on TN and soil microbial communities (Fig. 6b). The model fitted the data well (AIC = 76.94, *Fisher's C* = 26.94, *P* = 0.80). The results showed that nitrite respiration had a negative effect on TN content. TN indirectly affected the bacterial entire taxa by negatively affecting the bacterial rare taxa. Furthermore, TN indirectly affected the fungal entire taxa by positively affecting the fungal abundant taxa.

## DISCUSSION

Soil microorganisms are key indicators of soil quality. There are differences in the structure and function of the soil microbial community after the application of locust frass compared to chemical fertilizer. Firstly, the structure of microbial communities in the locust frass treatment and chemical fertilizer treatment differed significantly at the phylum level, genus level, and OTU level. Secondly, compared to chemical fertilizer, locust frass application enriched functional groups related to nitrogen cycle (especially after 2 years of treatment). The nitrogen present in gypsy moth frass could stimulate nitrogen fixation by soil microorganisms (47). In our study, this phenomenon could underlie the observed enrichment of nitrogen cycle-associated microorganisms following the application of locust frass. Thirdly, the application of locust frass increased the alpha diversity of soil microbial communities compared to chemical fertilizer (especially after 2 years of treatment). Similarly, some studies have found that the application of insect frass can increase the alpha diversity of soil bacterial and fungal communities (14, 15). In general, the higher the soil microbial diversity, the higher the soil quality (48). Finally, locust frass application enhanced the network complexity of the soil microbial community (except for the bacterial community of the 2-year treatment). Some studies have found that the application of organic fertilizers formed more complex co-occurrence networks (49–51). The complexity of soil microbial community networks has been shown to maintain ecosystem function (52). Overall, soils applied with locust frass were healthier than those applied with chemical fertilizer in terms of soil microbial community composition.

Rare taxa play an important role in microbial communities, such as maintaining community diversity (53), transmitting co-occurrence network information (54), and responding to environmental changes (55). In this study, the effects of different fertilization treatments on the soil microbial community responded mainly to rare taxa. Firstly, after two or three consecutive years of the application of locust frass and chemical fertilizer, only rare taxa were significantly segregated (Fig. 2b). This is probably because rare taxa are the species pool that maintain community diversity (53). Secondly, the nodality and proximity centrality of the bacterial and fungal rare taxa were significantly higher than the abundant taxa, suggesting that more information of the network may be transmitted through rare taxa (54). Finally, the bacterial rare taxa under the locust frass treatment are more correlated with environmental factors. This may be due to the fact that rare taxa are more susceptible to perturbation by environmental factors (55). Chen et al. (56) found that fertilization can alter soil multifunctionality by affecting rare taxa without altering abundant taxa. This may imply that the alteration of the bacterial rare taxa by locust frass application in this study will eventually respond to the multifunctionality of the soil.

Keystone microbes have a pivotal influence on the structure and function of microbial communities (57). The bacterial community structure varied more after 2 years of treatments than after 3 years, while the fungal community structure was relatively stable after both 2 and 3 years of treatment (Fig. 1c, 2b, 3). Based on the results of two

consecutive years of treatment, we hypothesize that there are distinct keystone microbes associated with locust frass and chemical fertilizer treatments. Random forest machine learning models were applied to find *Bacillus* as keystone microbes that discriminate among different treatments after two consecutive years across the bacterial entire, abundant, and rare taxa. *Bacillus* is well known as a genus of beneficial bacteria that have various beneficial impacts on plant growth (58). The growth-promoting potential of *Bacillus* isolated from the rhizosphere soil of the peach trees in our study was assessed, and the pot results showed that all three *Bacillus* strains could promote plant growth. However, *Bacillus* abundance increased after 2 years of chemical fertilizer treatment but significantly declined after 3 years, which implied that the continuous application of chemical fertilizer may negatively affect the abundance of potential growth-promoting bacteria.

An interesting question is whether changes in the soil microbial community are caused by microbes colonizing the soil in insect frass. Fuhrmann et al. (59) revealed a marked divergence in the rhizosphere soil microbial community post-application of both sterilized and non-sterilized black soldier fly frass. Consequently, it was inferred that the microorganisms intrinsic to the insect frass could exert a significant impact on the soil microbial community. Nonetheless, these authors surmised that alterations in the nutrient profile of the fertilizer due to sterilization might influence the soil microbial community. In our study, a microbial source-tracking method was used to ascertain whether the microorganisms indigenous to locust frass have the potential to colonize the soil and thereby influence the soil microbial community. Results indicated that a mere fraction, less than 0.6%, of the soil microbial community consisted of microorganisms from locust frass, while the overwhelming majority, approximately 90%, was soil derived. Thus, it was deduced that the microorganisms from locust frass were unlikely to establish long-term colonization within the soil. The potential impact of ephemeral colonization by microorganisms from locust frass on soil microbial communities cannot be conclusively dismissed. Further inquiry is warranted to elucidate the influence of the microorganisms in insect frass on the soil microbial community.

Soil physicochemical parameters are another key indicator of soil quality. The application of locust frass and chemical fertilizer showed different effects on soil properties. The continuous application of chemical fertilizer for 3 years resulted in a significant decrease in the TN, AN, and AP contents, compared to the application for only two consecutive years. In contrast, there were no significant changes in soil physicochemical properties and enzyme activities after 2 or 3 years of continuous application of locust frass, indicating that the effects of locust frass on soil quality were stable with increasing treatment duration. Contrary to expectations, soil nitrogen content decreased with increasing application time under chemical fertilizer treatments. However, this phenomenon of nitrogen loss has been confirmed in some studies (60–62). Based on 15N-tracer studies, 10% to 50% of the applied nitrogen was immobilized by microbes within the first year after fertilization, but nitrogen losses did occur with increasing application time (63, 64). In this study, the genus *Bacillus*, which includes some nitrogen-fixing strains (65), was the keystone microbes in the soil after 2 years of continuous application of chemical fertilizer. However, the abundance of *Bacillus* decreased after 3 years of continuous application, which might partially explain the soil nitrogen content loss with increasing application time. After two consecutive years of application, there was no significant difference in AN between locust frass and chemical fertilizer. After three consecutive years of application, the AN content was significantly higher in soils with locust frass than in soils with chemical fertilizer. The enhancement of soil nitrogen content by insect frass has been demonstrated in many studies, and it is the most extensively investigated mechanism for its use as a fertilizer (16). Locust frass contains substantial nitrogen that can be transformed into ammonium and nitrate nitrogen (66). Nitrogen from insect frass can be indirectly transferred to plants through bacterial nitrogen metabolism (67). The locust frass application enhanced the abundance of some functional groups related to the nitrogen cycle in the soil after two

consecutive years of application, compared to the chemical fertilizer treatment. Among nitrogen metabolism-related functional group enriched in the locust frass treatment, nitrite respiration was significantly correlated with TN. Despite the lack of significant variation in the TN content across years in the locust frass treatments, TN still significantly influenced soil bacterial and fungal communities. The SEM results showed that TN indirectly affects the bacterial and fungal entire taxa by directly affecting the bacterial rare taxa and fungal abundant taxa, respectively. Furthermore, plant uptake of nitrogen could contribute to the observed diminution in soil nitrogen contents. The application of insect frass reduced the carbon-to-nitrogen ratio within plant tissue (14). This phenomenon might also elucidate the progressive depletion of soil nitrogen content with the extended duration of locust frass application delineated in this research. Similar to the increased TN and AN content, the elevated AP content after 2 years of continuous chemical fertilizer application might be associated with the abundance of *Bacillus*. Some *Bacillus* species have been reported to be involved in phosphate dissolution, such as *B. megaterium* (68). SEM revealed that the *Bacillus* in the bacterial rare taxa had a positive influence on the AP content, while the AP content negatively affected the bacterial abundant and rare taxa. Among the various enzyme activities of the soil, only CAT was significantly higher in locust-frass-treated soil after 2 years of continuous application than in chemical-fertilizer-treated soil. CAT activity is closely associated with the population of aerobic microorganisms and the redox capacity of the soil (69). The application of locust frass increased soil CAT activity, thereby reducing the toxic effects of hydrogen peroxide on soil microorganisms. Except for CAT, soil enzyme activities were not significantly different between the two treatments.

There are several factors that constrained our research. We concentrated on the changes in soil bacterial and fungal communities following the locust frass application. More studies indicated that insect frass application engendered changes in the soil's bacterial and fungal community compositions, uncovering multifarious functions of these microorganisms, such as nitrogen metabolism, chitin degradation, and organic phosphorus mineralization (14, 15). Furthermore, archaea play an important role in the soil nitrogen cycle and could potentially be significant following the application of insect frass (13, 70). Our future research will include a detailed examination of archaea. Some studies indicated that the application of insect frass could contribute to an elevation in fungal biomass (59, 71–73). The high-throughput sequencing technique used in this study could not measure biomass, and our primary interest was in observing changes in the structure and diversity of the microbial community. In this study, functional inference of bacterial communities based on high-throughput sequencing was used to resolve functional groups enriched in soil after fertilizer application. However, the results were limited by horizontal gene transfer, database coverage, and accuracy (74). In future studies, metagenomic sequencing and other methods could be used to deeply resolve the functions of soil microbial communities. Differences in soil properties and microbial community structure observed among the different treatments were based on the quantities of locust frass and chemical fertilizer applied in our study. It should be noted that modifications to these application quantities could potentially alter the resultant findings. In addition, our study was limited by a short fertilization period, and long-term applications should be conducted in the future, which may provide more accurate evidence on the development and utilization of locust frass.

## Conclusion

This study investigated the effects of locust frass on soil properties and microbial communities in a peach orchard. The results showed that locust frass application increased the diversity and complexity of the soil microbial community, especially the rare taxa, compared to chemical fertilizer. Additionally, locust frass enhanced the abundance of functional groups related to nitrogen cycle and the contents of nitrogen. *Bacillus* was identified as keystone microbes that discriminated between locust frass and chemical fertilizer treatments, and showed plant-promoting properties in

pot experiments. Moreover, *Bacillus* had a positive effect on the AP content. However, the abundance of *Bacillus* was decreased with increasing chemical fertilizer treatment duration. In soils treated with chemical fertilizer, available phosphorus emerged as the only physicochemical parameter exhibiting a significant association with microbial populations. Conversely, in soils amended with locust frass, total nitrogen demonstrated the most substantial correlation with soil microbial communities. Overall, the study suggests that locust frass showed some positive effects on soil nutrients and the soil microbial community, particularly on rare taxa, contributing to sustainable agriculture development.

## ACKNOWLEDGMENTS

This work was supported by the National Natural Science Foundation of China (42077027 and 42377309), the Natural Science Foundation of Shandong Province (ZR2022MC198), the Science and Technology of Small and Medium Enterprises Innovation Ability Enhancement in Shandong Province (2023TSGC0345), and the Key R&D Plan Project in Ningxia Hui Autonomous Region (2021BBF02006).

The authors declare that they have no known competing financial interests or personal relationships that could have appeared to influence the work reported in this paper.

## AUTHOR AFFILIATIONS

[1]College of Life Sciences, Shandong Agricultural University, Tai'an, Shandong, China
[2]College of Plant Protection, Shandong Agricultural University, Tai'an, Shandong, China
[3]Shandong Agriculture and Engineering University, Jinan, Shandong, China

## AUTHOR ORCIDs

Zheng Gao  http://orcid.org/0000-0003-1691-0976
Ningxin Wang  http://orcid.org/0000-0002-7058-4959

## FUNDING

| Funder | Grant(s) | Author(s) |
| --- | --- | --- |
| National Natural Science Foundation of China | 42077027, 42377309 | Zheng Gao |
| Natural Science Foundation of Shandong Province | ZR2022MC198 | Ningxin Wang |
| Science and Technology of Small and Medium Enterprise Innovation Ability Enhancement in Shandong Province | 2023TSGC0345 | Ningxin Wang |
| Key R&D Plan Project in Ningxia Hui Autonomous Region | 2021BBF02006 | Ningxin Wang |

## AUTHOR CONTRIBUTIONS

Ji'ang Nie, Data curation, Writing – original draft | Hao Chen, Investigation, Methodology, Resources | Yongdong Wang, Data curation, Resources | Dapeng Zhang, Methodology, Resources | Yao Wang, Methodology, Resources | Ningxin Wang, Conceptualization, Funding acquisition, Investigation, Writing – review and editing.

## DATA AVAILABILITY

Bacterial and fungal raw reads were stored in the NCBI database with the accession numbers PRJNA1017155 and PRJNA1017376, respectively. Additionally, bacterial and fungal raw reads used to track the potential sources were stored in the NCBI database with the accession numbers PRJNA1045318 and PRJNA1045327, respec-

tively. The supporting information is also available at https://doi.org/10.6084/m9.figshare.28471364.

## ETHICS APPROVAL

This article does not contain any studies with human participants or animals performed by any of the authors.

## ADDITIONAL FILES

The following material is available online.

### Open Peer Review

**PEER REVIEW HISTORY (review-history.pdf).** An accounting of the reviewer comments and feedback.

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
