## [Reviewer comments · Microbiology Spectrum]

Microbiology Spectrum

Effects of applying locust frass on the soil properties and microbial community in a peach orchard

Ji'Ang Nie, Hao Chen, Yongdong Wang, Dapeng Zhang, Yao Wang, Zheng Gao, and Ningxin Wang

Corresponding Author(s): Ningxin Wang, Shandong Agricultural University

Review Timeline:

Submission Date:	October 28, 2024
Editorial Decision:	December 11, 2024
Revision Received:	December 23, 2024
Editorial Decision:	January 21, 2025
Revision Received:	January 30, 2025
Accepted:	January 31, 2025

Editor: Charina Gracia Banaay

Reviewer(s): Disclosure of reviewer identity is with reference to reviewer comments included in decision letter(s). The following individuals involved in review of your submission have agreed to reveal their identity: Laith Khalil Tawfeeq Al-Ani (Reviewer #1)

Transaction Report:

DOI: <https://doi.org/10.1128/spectrum.02470-24>

Re: Spectrum02470-24 (Effects of applying locust frass on the soil properties and microbial community in a peach orchard)

Dear Prof. Ningxin Wang:

Thank you for the privilege of reviewing your work. Below you will find my comments, instructions from the Spectrum editorial office, and the reviewer comments.

Revision Guidelines

Sincerely,
Charina Gracia Banaay
Editor
Microbiology Spectrum

Reviewer #1 (Comments for the Author):

I don't find any change in keywords
I mentioned to authors to change keywords

Reviewer #2 (Comments for the Author):

The manuscript entitled "Effects of applying locust frass on the soil properties and microbial community in a peach orchard" provided an engineering view about the microbial community variation in response to the addition of insect frass rather than chemical fertilizers. The results would be meaningful to guide the application of biofertilizers. However, the fundamental data would be a limitation to confirm if the results would be extensively feasible, as lack of long-term monitoring data and sufficient replicates.

1 In the method section, authors should clarify how many bio-samples collected from the CF and LF area were transferred for DNA extraction and following analysis.

2 The addition of locust frass has operated for several years since 2016. The time series data would be meaningful to show the effect of animal manure compared with chemical fertilizers. Long-term monitoring data would be greatly appreciated.

3 How many replicates for the CF and LF samples, only two samples were shown in Figure 1, it would be arbitrary to confirm the difference between CF and LF in terms of bacteria and fungi community.

4 Any references for the threshold of abundant and rare taxa?

5 What are the parameters for the microbial network construction? For example, how many replicates, which correlation method, etc.

Dear editors and reviewers:

On behalf of all the contributing authors, we would like to express our sincere appreciation for your constructive suggestions on our article “Effects of applying locust frass on the soil properties and microbial community in a peach orchard” (Manuscript Number Spectrum02470-24).

We carefully read your response letter and the reviewers’ comments. We highly appreciate the critical comments given for our manuscript. These suggestions were seriously discussed, and we revised our manuscript according to the suggestions. Point-by-point responses to the reviewers’ suggestions are listed as follows. Thank you so much for your help.

Looking forward to your reply.

Yours sincerely,

Ningxin Wang

College of Plant Protection, Shandong Agricultural University, Tai’an, Shandong,
271018, China

E-mail: nxwang@sdau.edu.cn

Reviewer #1 (Comments for the Author):

I don't find any change in keywords

I mentioned to authors to change keywords

Response: Thank you very much for your comment. We replaced “Insect frass” and “Peach” with “*Locusta migratoria*” and “Peach orchard” respectively in our revised manuscript. (Lines 44-45)

Reviewer #2 (Comments for the Author):

The manuscript entitled "Effects of applying locust frass on the soil properties and microbial community in a peach orchard" provided an engineering view about the microbial community variation in response to the addition of insect frass rather than chemical fertilizers. The results would be meaningful to guide the application of biofertilizers. However, the fundamental data would be a limitation to confirm if the results would be extensively feasible, as lack of long-term monitoring data and sufficient replicates.

Response: We show great thanks for your professional review work on our manuscript. According to your professional suggestions, we made corrections in our revised manuscript.

1 In the method section, authors should clarify how many bio-samples collected from the CF and LF area were transferred for DNA extraction and following analysis.

Response: Thank you very much for your comment. We added the number of bio-samples in the method section. Here are the modifications:

Five soil samples were collected as five replicates for subsequent analysis and microbial DNA extraction. (Lines 137-138)

2 The addition of locust frass has operated for several years since 2016. The time series data would be meaningful to show the effect of animal manure compared with

chemical fertilizers. Long-term monitoring data would be greatly appreciated.

Response: Thank you very much for your professional guidance. We conducted fertilizer experiments in peach orchards from 2016 to 2019 to observe the effects of long-term applications of locust frass on peach trees. We also wanted to do the long-term monitoring, unfortunately, our fertilization experiment was interrupted by the epidemic in 2020-2021. For the sake of accuracy, we only analyzed the data from these years.

3 How many replicates for the CF and LF samples, only two samples were shown in Figure 1, it would be arbitrary to confirm the difference between CF and LF in terms of bacteria and fungi community.

Response: Thank you very much for your comment. Our previous description of Figure 1 was unclear. All data in Figure 1 are based on five replicates per treatment. The bar graphs show the mean values for each treatment. Here are the modifications in Figure 1:

The top 10 phyla and genera with the highest average relative abundance for each treatment were presented.

4 Any references for the threshold of abundant and rare taxa?

Response: Thank you very much for your comment. The threshold of abundant and rare taxa in our manuscript are informed by the following two references.

35. Jiao S, Lu Y. 2020. Abundant fungi adapt to broader environmental gradients than rare fungi in agricultural fields. *Global Change Biology* 26:4506-4520. <https://doi.org/10.1111/gcb.15130> (Lines 766-768)

36. Zhao Z, Ma Y, Feng T, Kong X, Wang Z, Zheng W, Zhai B. 2022. Assembly processes of abundant and rare microbial communities in orchard soil under a cover crop at different periods. *Geoderma* 406:115543. <https://doi.org/10.1016/j.geoderma.2021.115543> (Lines 769-772)

5 What are the parameters for the microbial network construction? For example, how

many replicates, which correlation method, etc.

Response: Thank you for your suggestion. We added the parameters for the microbial network construction in our revised manuscript. Here are the modifications:

Associations were determined by Spearman's correlation coefficient. Associations with $P < 0.05$ and correlation coefficients above 0.4 were retained. (Lines 247-249)

And we added a characterization of the number of replicates in Figure 3.

Re: Spectrum02470-24R1 (Effects of applying locust frass on the soil properties and microbial community in a peach orchard)

Dear Prof. Ningxin Wang:

Thank you for the privilege of reviewing your work. Below you will find my comments, instructions from the Spectrum editorial office, and the reviewer comments.

Thank you for addressing the reviewer's comments adequately. As a final review, please go through the following minor comments and suggestions.

Line 190-191 - Revise statement to read "Sequence homology using BLAST in the NCBI database (<https://www.ncbi.nlm.nih.gov/>) was done for the obtained 16SrRNA gene sequences.

Line 197 - Where were these strains isolated from? Please indicate that the 3 bacterial strains were isolated from the peach tree rhizosphere as described in the previous section

Line 235 - revise to "while relative abundances less than 0.01% were defined asw "rare taxa""

Line 242-243 - revise to "...using the Shapiro-Wilk normality test in the R package MVN."

Line 244 - revise Blast to BLAST

Lines 295-297 - Revise to "There were 32 and 14 bacterial and fungal OTUs, respectively, that differed in both the two-and three-year treatments."

Lines 307-309 - For greater clarity, please revise to... "Though the nitrogen metabolism functional groups seemed to be numerically higher in the LF than the CF treatment, the difference was not statistically significant."

Lines 319-320 - the phrase "...while were not significant in all years and taxa" is confusing, please clarify what you mean

Lines 323-326 - For greater clarity, please revise to... "No statistically significant differences were observed for the Shannon diversity index and the Pielou's evenness index computed for CF and LF."

Line 465 - "indicator" should be plural "indicators"

Lines 484-485 - revise the statement to "...in terms of soil microbial community composition"

Line 487 - Briefly state examples of these roles

Line 582 - What is the significance of finding that CAT was significantly higher in the locust frass treatment?

Revision Guidelines

Sincerely,
Charina Gracia Banaay
Editor
Microbiology Spectrum

Dear editors and reviewers:

On behalf of all the contributing authors, we would like to express our sincere appreciation for your constructive suggestions on our article “Effects of applying locust frass on the soil properties and microbial community in a peach orchard” (Manuscript Number Spectrum02470-24R1).

We carefully read your response letter. We highly appreciate the minor comments given for our manuscript. These suggestions were seriously discussed, and we revised our manuscript according to the suggestions. Point-by-point responses to the reviewers’ suggestions are listed as follows. Thank you so much for your help.

Looking forward to your reply.

Yours sincerely,

Ningxin Wang

College of Plant Protection, Shandong Agricultural University, Tai’an, Shandong,
271018, China

E-mail: nxwang@sdau.edu.cn

Line 190-191 - Revise statement to read "Sequence homology using BLAST in the NCBI database (<https://www.ncbi.nlm.nih.gov/>) was done for the obtained 16SrRNA gene sequences.

Response: Thank you very much for your comment. We revised this sentence in our revised manuscript. Here are the modifications:

Sequence homology using BLAST in the NCBI database (<https://www.ncbi.nlm.nih.gov/>) was done for the obtained 16SrRNA gene sequences. (Lines 186-188)

Line 197 - Where were these strains isolated from? Please indicate that the 3 bacterial strains were isolated from the peach tree rhizosphere as described in the previous section

Response: Thank you for your comment. We supplemented this sentence with information indicating that these three strains were isolated from the peach tree rhizosphere. Here are the modifications:

Considering the perennial nature and long growth cycle of peach tree, we selected the common vegetable crop Chinese cabbage (*Brassica campestris* L.) as a suitable host for testing three bacteria strains isolated from the peach tree rhizosphere and conducting plant growth experiments. (Lines 192-195)

Line 235 - revise to "while relative abundances less than 0.01% were defined as "rare taxa""

Response: Thank you for your comment. We revised this sentence in our revised manuscript. Here are the modifications:

When the relative abundance of OTUs in all samples was more than 0.1%, they were defined as "abundant taxa", while the relative abundances was less than 0.01%, they were defined as "rare taxa". (Lines 230-232)

Line 242-243 - revise to "...using the Shapiro-Wilk normality test in the R package MVN."

Response: Thank you for your comment. We revised this sentence in our revised manuscript. Here are the modifications:

The normality of the data was assessed using the Shapiro-Wilk normality test in the R package “MVN”. (Lines 238-239)

Line 244 - revise Blast to BLAST

Response: Thanks for your careful checks. We revised this error in our revised manuscript. Here are the modifications:

BLAST was used to compare the sequences of isolated *Bacillus* and OTUs. (Lines 240-241)

Lines 295-297 - Revise to "There were 32 and 14 bacterial and fungal OTUs, respectively, that differed in both the two-and three-year treatments."

Response: Thank you for your comment. We revised this sentence in our revised manuscript. Here are the modifications:

There were 32 and 14 differential bacterial and fungal OTUs, respectively, that differed in both the two- and three-year treatments. (Lines 290-291)

Lines 307-309 - For greater clarity, please revise to... "Though the nitrogen metabolism functional groups seemed to be numerically higher in the LF than the CF treatment, the difference was not statistically significant."

Response: Thank you for your comment. We revised this sentence in our revised manuscript. Here are the modifications:

Though the nitrogen metabolism functional groups seemed to be numerically higher in the LF than the CF treatment, the difference was not statistically significant. (Lines 301-303)

Lines 319-320 - the phrase "...while were not significant in all years and taxa" is confusing, please clarify what you mean

Response: Thanks for your careful checks. We revised this sentence in our revised

manuscript. Here are the modifications:

The Shannon index and Pielou's evenness index for both bacterial and fungal communities in the LF treatments were higher than in the CF treatments, but the differences were not statistically significant after three years of continuous application (Fig. 2a). (Lines 312-315)

Lines 323-326 - For greater clarity, please revise to... "No statistically significant differences were observed for the Shannon diversity index and the Pielou's evenness index computed for CF and LF."

Response: Thank you for your comment. We revised this sentence in our revised manuscript. Here are the modifications:

After three years of continuous application, no statistically significant differences were observed for the Shannon diversity index and the Pielou's evenness index computed for CF and LF. (Lines 318-320)

Line 465 - "indicator" should be plural "indicators"

Response: Thanks for your careful checks. We revised this error in our revised manuscript. Here are the modifications:

Soil microorganisms are key indicators of soil quality. (Lines 457)

Lines 484-485 - revise the statement to "...in terms of soil microbial community composition"

Response: Thank you for your comment. We revised this sentence in our revised manuscript. Here are the modifications:

Overall, soils applied with locust frass was healthier than chemical fertilizer in terms of soil microbial community composition. (Lines 475-477)

Line 487 - Briefly state examples of these roles

Response: Thank you very much for your professional guidance. We add the important role that rare taxa play in microbial communities. Here are the

modifications:

Rare taxa play an important role in microbial communities, such as maintaining community diversity (53), transmitting co-occurrence network information (54), and responding to environmental changes (55). (Lines 479-481)

Line 582 - What is the significance of finding that CAT was significantly higher in the locust frass treatment?

Response: We sincerely appreciate your valuable comments. We added the significance of elevated CAT activity after application of locust frass in the revised manuscript. Here are the modifications:

CAT activity is closely associated with the population of aerobic microorganisms and the redox capacity of the soil (69). The application of locust frass increased soil CAT activity, thereby reducing the toxic effects of hydrogen peroxide on soil microorganisms. (Lines 576-579)

Re: Spectrum02470-24R2 (Effects of applying locust frass on the soil properties and microbial community in a peach orchard)

Dear Prof. Ningxin Wang:

Your manuscript has been accepted, and I am forwarding it to the ASM production staff for publication. Your paper will first be checked to make sure all elements meet the technical requirements. ASM staff will contact you if anything needs to be revised before copyediting and production can begin. Otherwise, you will be notified when your proofs are ready to be viewed.

Sincerely,
Charina Gracia Banaay
Editor
Microbiology Spectrum